# LIRA: A Learning-based Query-aware Partition Framework for Large-scale ANN Search

## Abstract

Approximate nearest neighbor (ANN) search is fundamental in various applications such as information retrieval. To enhance efficiency, partition-based methods are proposed to narrow the search space by probing partial partitions, yet they face two common issues. First, in the query phase, a widely adopted strategy in existing studies such as IVF is to probe partitions based on the distance ranks of a query to partition centroids. This inevitably leads to irrelevant partition probing, since data distribution is not considered. Second, in the partition construction phase, all the partition-based methods have the boundary problem that separates a query's $k$NN to multiple partitions and produces a long-tailed $k$NN distribution, degrading the optimal $nprobe$ (i.e., the number of probing partitions) and the search efficiency. To address these problems, we propose *LIRA*, a LearnIng-based queRy-aware pArtition framework. Specifically, we propose a probing model to learn and directly probe the partitions containing the $k$NN of a query. Probing partitions with the model can reduce probing waste and allow for query-aware probing with query-specific $nprobe$. Moreover, we incorporate the probing model into a learning-based redundancy strategy to mitigate the adverse impact of the long-tailed $k$NN distribution on partition probing. Extensive experiments on real-world vector datasets demonstrate the superiority of *LIRA* in the trade-off among accuracy, latency, and query fan-out. The results show that *LIRA* consistently reduces the latency and the query fan-out up to 30%. The codes are available at https://anonymous.4open.science/r/Web-conference-1603/README.md.

## Keywords

Approximate nearest neighbor search, Learning-to-index

**ACM Reference Format:**
Anonymous Author(s). 2024. LIRA: A Learning-based Query-aware Partition Framework for Large-scale ANN Search. In . ACM, New York, NY, USA, 12 pages. https://doi.org/10.1145/nnnnnnn.nnnnnnn

## 1 Introduction

The nearest neighbor (NN) search is well studied in the community of information retrieval [30, 38, 42, 47, 48]. By encoding unstructured data (e.g., texts and images) into vectors with embedding models [23, 33], the semantic similarity of unstructured data can be represented as the similarity of vectors [39, 41]. Hence, NN search in the vector space is fundamental for efficiently retrieving

large-scale unstructured data in many applications [20, 36] such as Large Language Models (LLM) and Retrieval-Augmented Generation (RAG) [5, 11, 37]. Given a dataset $\mathcal{D}$ consisting $N$ vectors and a query vector $q$ in the same vector space, the goal of $k$NN search is to find the $k$ vectors nearest to $q$ from the dataset. However, with the growth of dataset cardinalities and dimensions, the *exact* NN search is too time-consuming to meet latency demands in practice [2, 18]. Consequently, current works shift focus towards approximate nearest neighbor (ANN) search [4, 26], which seeks a trade-off between the acceptable latency and desired accuracy by retrieving with indexing techniques.

### 1.1 Prior Approaches and Limitations

Partition-based methods are the backbone of ANN search in industry [5, 11, 34], which are suitable in disk-based and distributed scenarios through partial data loading [10, 15]. For example, rather than build one index on the whole dataset, Zilliz [34] separates the dataset into partitions and builds an individual index for each partition. However, there is a trade-off between fewer probing partitions and high recall. In detail, the $k$ nearest neighbors ($k$NN) of a query $q$ can be separated into several partitions. We denote the partitions containing the $k$NN of query $q$ as its **$k$NN partitions**, which means these partitions should be probed to retrieve the exact top-k nearest neighbors. A lower **$nprobe$** (i.e., the number of probing partitions, also known as the query fan-out) is preferred in partition-based scenarios for contributing to higher scalability [31, 44]. A naive way to achieve a low $nprobe$ can be partition pruning, but a trade-off exists between the effectiveness and efficiency of a pruning method. If the probing partitions do not cover the $k$NN partitions well, the recall of retrieval results will degenerate. If some probing partitions are not in the $k$NN partitions, there will be a waste of searching partitions without $k$NN. We call this phenomenon the **Curse of Partition Pruning**.

Here, we go through some partition-based ANN search methods and illustrate their limitations of partition pruning. Considering the low accuracy of tree-based [7, 13] and hash-based [32, 35, 38] methods, clustering methods are promising in dealing with the dilemma of partition pruning. The inverted file (IVF) index [18] builds clusters with the K-Means algorithm and then searches in fixed $nprobe$ nearest partitions **according to the distance rank between a query and the cluster centroids**. Further, BLISS [12] build partitions with deep learning models but still set a fixed $nprobe$ for all queries to achieve the trade-off between search latency and recall. However, since no fixed $nprobe$ can fit all queries [46], there remain limitations in search performance, such as the query fan-out for probing partitions and the search latency.

**Limit 1. Partition pruning with the distance rank to partition centroids wastes** $nprobe$. As shown in a toy example in Fig. 1, suppose the data points of a dataset are divided into partitions. The top 10 NNs of a query $q$ are distributed in three partitions (i.e., partition A, B, and C), which are ranked as the nearest, the second

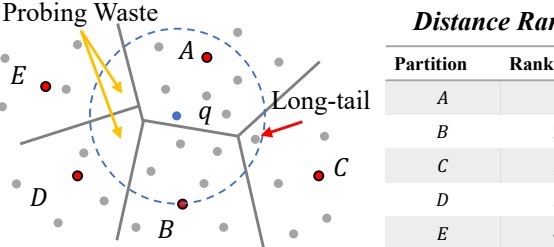

**Figure 1: Example for probing waste. The blue point is a query, and the red points are the centroids of partitions. A lower distance rank means a nearer centroid to the query.**

nearest, and the fifth nearest partition of $q$, respectively. To ensure all 10 NNs are included, the minimum number of probing partitions based on the centroid distance rank is $nprobe = 5$. Alternatively, a more cost-effective way is to directly probe the three $k$NN partitions, resulting in the optimal $nprobe$ to 3. Hence, pruning partitions according to the centroid distance rank still wastes $nprobe$, and such probing waste is ubiquitous in high-dimension datasets as illustrated in Section 2.

**Limit 2. Hard partitioning cannot inherently achieve low $nprobe$ due to the long-tailed distribution of $k$NN.** Partitioning strategies aim to reduce the number of probing partitions, $nprobe$. However, due to the curse of dimension and local density variations, the $k$NN distribution with hard partitioning methods (i.e., each data point is put in one partition) often exhibits the notorious long-tailed characteristic. Specifically, while most $k$NN may be densely located in a few partitions, the remaining $k$NN scatter across many other partitions [14]. As the example shown in Fig. 1, consider a scenario where the top-10 $k$NN of a query $q$ are distributed as [5, 4, 1, 0, 0] in five partitions. The scattered one $k$NN can be regarded as the long-tail $k$NN in the whole $k$NN distribution, resulting in the search process less efficient. Limited by the long-tailed $k$NN distribution among one group of partitions, BLISS [12] and its variants [22] construct four independent indexes and search in four groups of partitions to achieve a high recall. Hence, the long-tailed distribution of $k$NN diminishes the cost-effectiveness of probing partitions and increases the query fan-out undesirably.

## 1.2 Our Solution

We find that the essential issue of the limitations mentioned above stems from the distribution of $k$NN in partitions. Hence, we improve the performance of partition-based ANN search from two aspects, query process and index construction.

**Insight 1. A meta index that directly probes the $k$NN partitions is required.** To avoid confusion, we define the index for inter-partitions that facilitates partition pruning as the **meta index**, and the index for intra-partition (i.e., the index that only organizes the data points within one partition) as **internal index**. In this study, we focus on the optimization of meta index among the two-level index for partitions, in which the internal index can apply any existing index structure such as HNSW [27]. As mentioned in Limit 1, the optimal number of probing partitions for a query $q$, denoted as $(nprobe^q)^*$, is exactly the number of its $k$NN partitions. When the context is clear, we refer to $(nprobe^q)^*$ as $nprobe^*$ in

this paper for brevity. To address the probing waste through an effective query process, the ideal meta index needs to directly probe the $k$NN partitions of a query. Compared to IVF, the meta index can achieve high recall while reducing the $nprobe$ simultaneously.

**Insight 2. Redundant partitioning is required to mitigate the long-tailed $k$NN distribution and further reduce $nprobe^*$.** As discussed in Limit 2, the search inefficiency often arises from the long-tailed $k$NN distribution. To address this from the aspects of index construction, a feasible approach is to redundantly put a query's long-tail $k$NN into other densely distributed $k$NN partitions. For instance, the initial optimal $nprobe$ of $q$ in Fig. 1 is $nprobe_q^* = 3$. By introducing redundancy - specifically, duplicating the single $k$NN in partition C to another $k$NN partition (i.e., partition A or B) - the revised optimal $nprobe$ of $q$ can be further reduced to $nprobe_q^* = 2$, since merely probing partitions A and B is enough to cover all the top-10 $k$NN of $q$. Hence, strategic redundancy can effectively reduce the number of probing partitions. However, instead of merely reducing the $nprobe$, the objective in partition-based ANN search is striking a better trade-off between latency and recall. As the number of data replicas increases, the $nprobe^*$ can be minimized to one, resulting in all data in a single partition but an increased search latency. Hence, we need an exquisite redundant partition method to balance partition pruning and data redundancy.

Based on the above insights, we propose *LIRA*, a LearnIng-based queRy-aware pArtition framework, which serves as a meta index across partitions. In general, *LIRA* follows the "one size does not fit all" principle and explores the power of adaption. After building initial partitions with K-Means, we utilize a learning model to tell us where to probe for individual queries dynamically, which data points need to be duplicated, and where to duplicate these data points adaptively across partitions. Specifically, we first enhance query process by developing a probing model to predict query-dependent $k$NN partitions, thereby enabling precise partition pruning with less probing waste. The probing model uses a tunable threshold on its output probabilities, allowing for more adaptive partition pruning and fine-grained tuning than the $nprobe$ configuration in IVF. Second, we improve the construction of partitions by efficiently duplicating data points with the same model. When building redundant partitions, we novelly transfer the task from an exhaustive search for $k$NN of all data points globally to discriminating data points individually. Finally, in the top-k retrieval phase, *LIRA* leverages the model's probing probabilities to guide the search process across partitions, reducing query fan-out and latency. To simulate the partition-based scenarios in practice, we combine the *LIRA* as the meta index with the HNSW index as the internal index. In summary, we make the following contributions.

- To save the probing waste in a query-specific way, we propose a learning-based partition pruning strategy where a probing model generates the probing probabilities of partitions for each query.
- To mitigate the effect of long-tailed $k$NN distribution by building redundant partitions, we novelly transfer the problem of duplicating data points globally to individually and then propose a learning-based redundancy strategy with the probing model.
- We conduct extensive experiments on publicly available high-dimensional vector datasets, demonstrating the superiority of *LIRA* in recall, latency, and partition pruning.

# 2 Preliminaries

## 2.1 Definitions

Suppose $\mathcal{D} = \{v^1, v^2, \ldots, v^N\}$ be a dataset of $N$ $d$-dimension data points separated in $B$ partitions, and $dist(v^1, v^2)$ is the function to calculate the distance between data points $v^1$ and $v^2$. We define the ANN search problem in the partition-based scenario as follows.

DEFINITION 1 (kNN COUNT DISTRIBUTION). *Given a query vector $q$, let $S_{GT} = \{v^1, v^2, \ldots, v^k\}$ be the ground truth (abbreviated as GT) set of $q$'s $k$ nearest neighbors. Let $n_i^q$ be the count of ground truth $k$NN of $q$ in the $i$-th partition, the $k$NN count distribution of a query $q$ can be defined as $n^q = [n_1^q, n_2^q, \ldots, n_B^q]$, where $\sum_{i=1}^{i=B} n_i^q = k$.*

DEFINITION 2 (RECALL@k). *Recall@k refers to the proportion of the ground truth top-k nearest neighbors retrieved by an ANN search method out of all ground truth $k$NN in the dataset.*

$$Recall@k = \frac{|S \cap S_{GT}|}{k} \times 100\%. \tag{1}$$

*where $S$ is the retrieved results. A higher Recall@k value indicates a greater number of exact top-k nearest neighbors are retrieved.*

DEFINITION 3 (LONG-TAIL DATA POINT). *Given a $k$NN count distribution of a query $q$, $n^q = [n_1^q, n_2^q, \ldots, n_B^q]$, we regard the part of $k$NN with $n_i^q = 1$ as the long-tail part in the $k$NN count distribution. The specific data point served as the long-tail $k$NN of $q$ where $n_i^q = 1$ in the long-tail part is termed as a long-tail data point.*

For a $k$NN count distribution $n^q$, we term the partition that contains the ground truth $k$NN as a **$k$NN partition**. We denote the $k$NN partition distribution $p^q = [p_1^q, p_2^q, \ldots, _B^q]$ as a binary mask over $k$NN count distribution, where $k$NN partitions are marked with 1 while others with 0, and $\sum_{i=1}^{i=B} p_i^q = (nprobe^q)^*$. In addition, for a long-tailed $k$NN count distribution $n^q$, we regard partitions with $n_i^q > 1$ as the **replica partition** (e.g., partition A and B in Fig. 1). According to Insight 2, duplicating a long-tail data point into its replica partitions can save one probing and reduce $nprobe^*$ further. There may be other replica partitions of a data point, since it can be in the long-tail part of the $k$NN distribution of other queries.

DEFINITION 4 (OBJECTIVE). *Compared with baseline methods under equivalent Recall@k, our objective is to optimize the partition pruning and query latency. Our approach involves the refinement of partition by integrating a learning-based redundancy strategy and query-aware retrieval process with a learned probing model.*

## 2.2 Motivation

We provide motivations through conducting preliminary studies on the SIFT [2] dataset with 1 million data points and 10K queries. First, we show the waste of probing cardinality, $nprobe$, caused by partition pruning with centroids distance ranks. Second, we illustrate that the long-tail $k$NN is common in $k$NN count distributions.

**Probing Waste with Distance Ranking.** In IVF, the probing cardinality $nprobe$ infers probing the nearest $nprobe$ partitions. Ideally, the optimal number of probing partitions, $nprobe^*$, should be no larger than $k$ for Recall@k. We define $nprobe_{dist}^*$ as the maximum distance rank among the $k$NN partitions, which implies the nearest $nprobe_{dist}^*$ partitions need to be probed to cover all $k$NN.

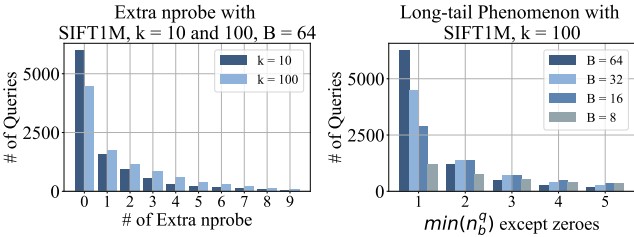

Figure 2: Extra probing with distance ranking (LEFT) and common phenomenon of long-tail $k$NN (RIGHT).

However, we find that probing according to $nprobe_{dist}^*$ wastes probing cardinality. For example, the $nprobe_{dist}^*$ exceeds 20 for some queries when retrieving top-10 $k$NN and the waste of probing is even worse with $k = 100$ (See detailed description in Appendix B). In addition, we show the extra $nprobe$ for each query when probing partitions according to the distance rank in Fig. 2 (LEFT). The extra $nprobe$ is the difference between the optimal $(nprobe^q)^*$ and the $(nprobe^q)_{dist}^*$. Hence, these observations suggest an opportunity to reduce the probing waste by refining the probing strategy during the query phase.

**Common Long-tail $k$NN.** To discover the ubiquity of long-tailed $k$NN count distribution, we analyze the long-tail phenomenon by calculating the minimum $n_i^q$ in a query $q$'s $k$NN count distribution expect zeros. Considering the $k$NN of a query can be more congested with larger partition sizes, we set the number of partitions $B \in \{64, 32, 16, 8\}$, respectively. As depicted in Fig. 2 (RIGHT), the long-tail phenomenon exists regardless of the number of partitions. In detail, as we can see in the horizontal axis with a value of 1, thousands of queries have long-tailed $k$NN count distribution among a total 10k queries. Hence, if valuable knowledge can be extracted from the $k$NN count distributions, there are two opportunities to improve the ANN search by incorporating redundancy during partition building. First, we can infer the latent long-tail data points among the whole dataset. Second, we can predict the replica partitions for long-tail data points. In Section. 3.3, we present the time complexity of building redundant partitions with global $k$NN count distributions of all data, and then introduce an efficient learning-based redundancy strategy.

# 3 Method

In this section, we first present an overview of *LIRA*. We then introduce the partition pruning strategy with a learned probing model. We illustrate the learning-based redundancy strategy by identifying long-tail data points and duplicating them to replica partitions with the probing model. Finally, we present $k$NN retrieval across partitions by using the probing model as the meta index.

## 3.1 Framework Overview

The workflow of *LIRA* can be divided into two distinct processes: the construction of redundant partitions and the top-k retrieval for queries, which we illustrate through a toy example with 5 partitions in Fig. 3 and 6, respectively. **(1) Probing model training.** After initializing the $B$ partitions with the vanilla K-Means algorithm,

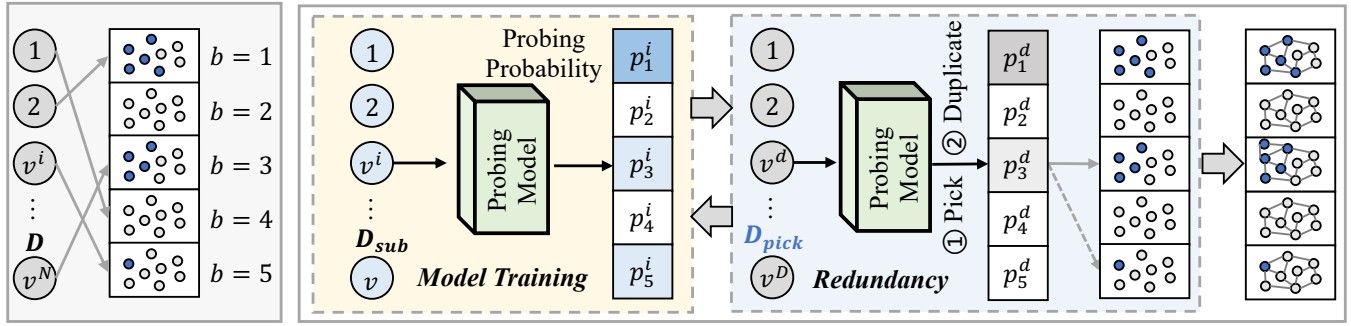

**Figure 3: Partition initialization (LEFT), the probing model training (MIDDLE) and learning-based redundancy strategy (RIGHT).**

*LIRA* targets training a probing model to serve as the meta index and to engage in building redundant partitions. Specifically, the probing model is applied threefold in *LIRA*: learning the mapping function $f(\cdot)$ of data points to $k$NN partitions in training, providing potential long-tail data points and replica partitions in learning-based redundancy, and guiding the partition probing during the query-aware retrieval process. **(2) Learning-based redundancy.** During the redundancy phase, we aim to duplicate the potential long-tailed data points to the replica partitions by using the probing model. Finding the deep correlation between the $k$NN partitions and the replica partitions, we novelly transform the problem of picking and duplicating long-tailed data points from globally to individually. First, we pick the data points with more predicted *nprobe* as the potential long-tail data points. Second, we choose the partition with a high predicted possibility in $\widehat{p^v}$ to put the replica of data point $v$. We construct the internal indexes for each partition individually after the redundant partitions are built (See detailed algorithm of the two-level index in Appendix C). **(3) Query-aware retrieval process.** Since the probing model can map a data point in the vector space to its $k$NN partitions, we can use it to guide the top-k retrieval across partitions. The predicted probabilities are utilized along with a probability threshold in the inter-partition pruning, and the internal indexes are used for inner-partition searching.

## 3.2 Probing Model Training

**Probing Model.** The model has two inputs: (1) the query vector $q$, and (2) the distances between $q$ and partitions centroids $I$. We can regard the probing model as a multivariate binary classifier for whether to probe each partition. The output is the predicted probability $\widehat{p}$ in the dimension of the number of partitions $B$. Hence, the model can be represented as $f(q, I) = \widehat{p}$. We convert the two inputs as feature vectors $x_q$ and $x_I$ with individual networks, respectively, and then concatenate the two feature vectors to generate the predicted probing probabilities $\widehat{p}$ as follows.

$$x_q = \phi_q(q), x_I = \phi_I(I), \widehat{p} = \phi_p(x_q \oplus x_I) \qquad (2)$$

where $\phi_q, \phi_I, \phi_p$ are three independent multi-layer models, and the output of $\phi_p$ is the predicted probabilities for probing partition $\widehat{p}$.

As discussed in Section 2.2, an ideal probing model served as the meta index should directly probe $k$NN partitions of a query regardless of its distance rankings to cluster centers. Hence, the labels of a $q$ are the same as the $k$NN partition distribution $p^q$, where partitions with $n_i^q > 0$ are regarded as positive and other partitions with no $k$NN are labeled as negative. For example, the labels of a $k$NN count distribution $[5, 4, 1, 0, 0]$ is $[1, 1, 1, 0, 0]$.

**Network Training.** We sample a subset of data from the whole dataset $D$ as training data and use the provided queries of a dataset to evaluate the effectiveness of *LIRA* (See detailed description on scalability in Appendix C and D.2). For each training data, we search the $k$NN from the training data to get the $k$NN partition distribution $p^q$. The output of the probing model $\widehat{p_b^q}$ in $[0, 1]$ is the possibility of probing each partition. We take the partition with $\widehat{p_b^q} \geq \sigma$ as a probing partition to support query-specific *nprobe*. The $\sigma$ is set as 0.5 in training and is tunable in the query process, which provides fine-grained tuning in partition pruning than the *nprobe* configuration in IVF. Hence, we can solve the multivariate binary classification problem with the cross-entropy loss:

$$\mathcal{L}(p^q, \widehat{p^q}) = -\sum_{b=1}^{B} \left( p_b^q \cdot \log(\widehat{p_b^q}) + (1 - p_b^q) \cdot \log(1 - \widehat{p_b^q}) \right) \qquad (3)$$

where the $\mathcal{L}(p^q, \widehat{p^q})$ is the loss of the probing model on a query $q$.

## 3.3 Learning-based Redundancy

Redundancy is a crucial step in *LIRA* since it helps refine the initial hard partitions as redundant partitions. The aim of redundancy is to reduce the side-effect of long-tailed $k$NN count distribution by reasonably duplicating each long-tail data point into one of its replica partitions. There are two important issues about redundancy. (1) Under the initial partition, how do we identify data points that tend to be the long-tail data points in the $k$NN count distribution of queries? (2) To duplicate the long-tail data points, which partitions should we transfer these data points into?

**Pick Data Points.** A data point $v$ might be a long-tail data point for any query. To identify whether a data point is long-tail, it is necessary to examine the $k$NN counts distribution, $n^q$, for all queries. Since the $n^q$ is unknown when building redundant partitions, we can only use the $k$NN count distribution of the data itself, $n^v$, which involves finding the data points in the long-tail $k$NN parts of other data's $k$NN count distribution. Duplicating globally means computing the $k$NN of all data to identify whether a data point is long-tail. However, the computation cost of getting $k$NN of the whole data

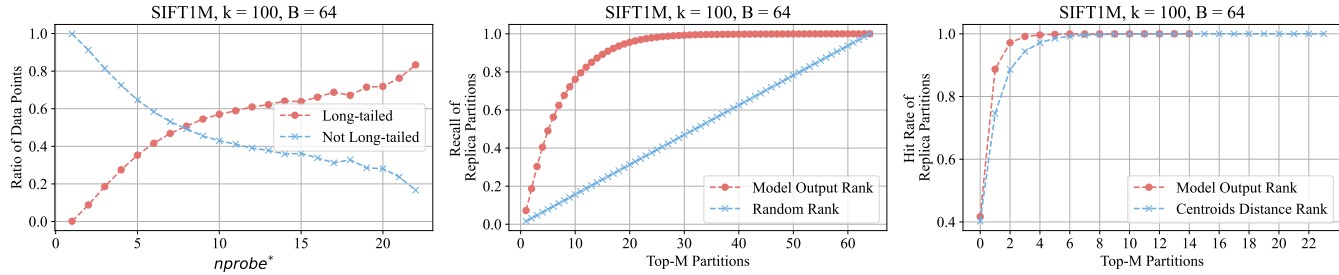

**Figure 4: Ratio of long-tailed and not long-tailed data points under certain $nprobe^*$ (LEFT). The recall (MIDDLE) and hit rate (RIGHT) of replica partitions among top-M partitions with model output rank or centroids distance rank.**

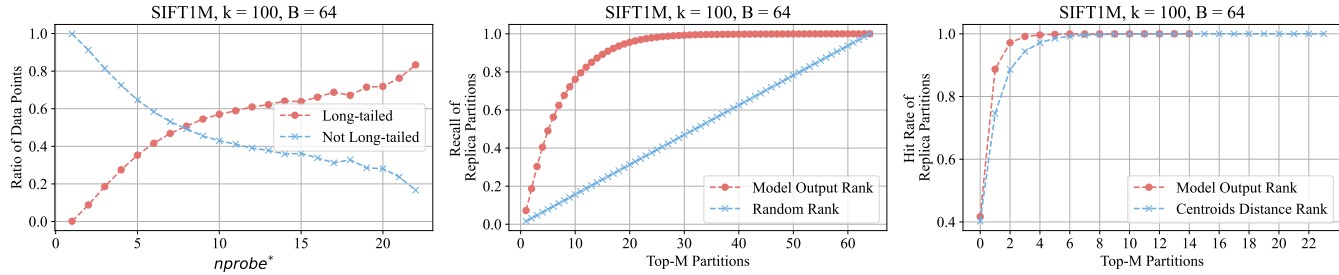

**Figure 5: Pick and duplicate potential long-tailed data points individually with the probing model is more efficient than using ground truth $k$NN count distribution globally.**

$O(N^2 \cdot d)$ is unacceptable in large-scale datasets. Consequently, it is impractical to identify all the long-tail data points globally. Innovatively, we circumvent this challenge and transfer the issue of identifying long-tail data points from globally to individually.

Specifically, we observe an interesting phenomenon from empirical analysis: **Data points with a larger $nprobe^*$ are more likely to be long-tail data points.** In detail, we record the $k$NN count distribution and $nprobe^*$ of individual data in SIFT and find the long-tail $k$NN data points. Varying the specific $nprobe^*$, we calculate the ratio of data that identified as long-tail data points versus those are not. As demonstrated in Fig. 4 (LEFT), an increase in $nprobe^*$ correlates with a higher ratio of long-tail data points. The observation aligns with the spatial partitioning in vector space: data points with $k$NN separated across multiple partitions are more likely located at the boundaries of partitions and thus are more prone to being long-tail.

In Fig. 5, we illustrate the transformation of picking data points by using other data's $k$NN globally to using ego $k$NN (i.e., the $k$NN of a data point itself) individually. For example, long-tail data points, $v^2$ and $v^3$, exhibit a higher $nprobe^*$ compared to the non-long-tail data point $v^1$. Leveraging the accurate prediction of the probing model, we can reliably use it to estimate the $nprobe^*$ of data points and pick potential long-tail data points individually. We apply the model to get the $\widehat{p}$ of all data points, selecting those within the upper

$\eta$ percentile of predicted $nprobe$. Hence, utilizing the probing model obviates the need to find $k$NN on whole data globally, streamlining the process of identifying long-tail data points.

**Duplicate Data Points.** After identifying data points requiring duplication, the next challenge is selecting appropriate partitions to put these replicas. Similar to the challenge of high computation cost in picking data points, the replica partitions of each long-tail data point are unknown if the global $k$NN count distribution is inaccessible. For all the long-tail data points, we record the $k$NN partitions and replica partitions, and we observe an interesting phenomenon: **the replica partitions for duplicating a data point $v$ have a strong relationship to its $k$NN partitions.** In detail, most of the replica partitions of $v$ align closely with its $k$NN partitions. As the example in duplicating data of Fig. 5, the $k$NN partition (i.e., the four partitions depicted in blue) of a long-tail data point, $v^2$, can cover its replica partitions (i.e., the three partitions shown in yellow). $v^1$ has no replica partitions since it is not a long-tail data point. This provides a promising approach to getting replica partitions by leveraging predicted probing partitions from the model.

Another problem follows this insight: since the model can produce many probing partitions for a data point $v$, which one should be chosen to put the replica of $v$? Our analysis indicates that **the partition $b$ with higher probing probability $p_b^v$ is more likely to be a replica partition for $v$.** In detail, we first get the replica partitions of all the long-tail data points. Then, we calculate $Recall_{rep}^v$, the recall between replica partitions and the top-$M$ predicted partitions, where $M$ ranges from 1 to $B$.

$$Recall_{rep}^v = \frac{|S_{model}^v \cap S_{rep}^v|}{|S_{rep}^v|} \times 100\%. \quad (4)$$

where $S_{model}^v$ is the set of top-M partitions in the model output of data point $v$, and $S_{rep}^v$ is $v$'s set of the replica partitions. For comparison, we also evaluate a random ranking of partitions represented by the blue line. (1) As shown in Fig. 4 (MIDDLE), we can see that the predicted probing partitions can effectively cover the replica partition, for $Recall_{rep}$ increases to nearly 1 with just $M = 20$. (2) In addition, the gradually decreasing slope of the red line can support that partitions with high output probabilities tend to be replica partitions. Hence, this insight informs our duplication strategy, which utilizes output probing probability. As illustrated in Fig. 5, if a long-tail data point is not in the partition with the highest output rank, we duplicate it into this partition; otherwise, we put it into the partition with the second-highest output rank.

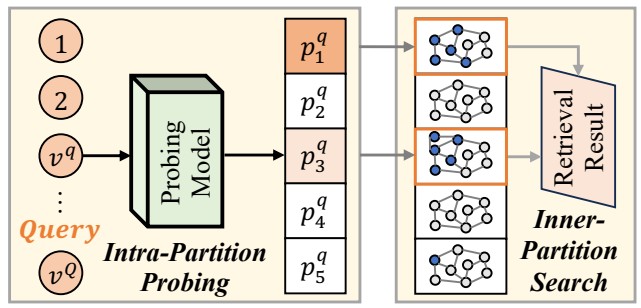

**Figure 6: Retrieval process across partitions.**

Furthermore, **compared with duplicating long-tail data points according to the distance rank of partition centroids, we observe that using the model output rank is more valid.** Typically, centroid distance ranks are considered when duplicating data points. For example, a data point can be duplicated up to 8 times in the closure clustering assignment in SPANN [5]. As we highlight in Limit 1, using centroid distance ranks often wastes *nprobe* for retrieving $k$NN. This limitation also emerges when choosing partitions to duplicate long-tail data points. For example, the long-tail data point $v^3$ in Fig. 5 has two replica partitions. The distance rank of replica partitions for $v^3$ is (2, 3), while the output rank can be (1, 2). We analyze the largest model output rank and centroid distance rank in replica partitions of each long-tailed data point, respectively. The hit rate with model output rank on data points $v$ is set to 1 if $|S_{model}^v \cap S_{rep}^v| \neq \emptyset$, otherwise it is set to 0. The hit rate with centroid distance rank is calculated similarly by using $S_{dist}^v$, the top-M partitions in the centroid distance ranking of $v$. As shown in Fig. 4 (RIGHT), the model output rank can better indicate the replica partitions for achieving a higher hit rate at the same $M$.

### 3.4 Query-aware Top-k Retrieval

After model training and learning-based redundancy, we can achieve retrieval with the meta index alone or with two-level indexes (i.e., the internal indexes are required in large-scale datasets) [43]. We store the probing model and redundant partitions to evaluate the performance of *LIRA* as a meta index for partitions. In the following part of this section, we give a detailed description of top-k retrieval with a two-level index and regard an exhaustive search in a partition if using meta index alone. As shown in Fig. 6, the retrieval process includes two stages. In the first stage, we utilize the probing model as the meta index to get the probing probabilities as the retrieval guidance. In the second stage, we execute the searching in each probing partition with internal indexes.

We illustrate the retrieval process with *LIRA* as the meta index and HNSW as the internal index as an example. (1) In the first stage, we utilize the probing model to obtain retrieval guidance. Similar to the training process, we apply the trained model to query vectors and get the probing probabilities *i.e.*, $\widehat{p^q}$. Instead of probing a fixed number of partitions, *LIRA* supports adaptive *nprobe* for each query with the predicted $\widehat{p^q}$, where only those partitions with $\widehat{p^q} > \sigma$ ($\sigma = 0.5$ for default) are treated as probing partitions. Hence, *LIRA* can prune more partitions and save more query fan-out compared

with a fixed *nprobe*. (2) In the second stage, *LIRA* executes the retrieval process with the probing partitions predicted in the first stage. When retrieving $k$ results in a probing partition for query $q$, we use the internal indexes without an exhaustive search in the partition. After completing searches in all probing partitions, we merge all the retrieved data points as a coarse candidate set. Then, we rank the coarse candidate set according to the distance to the query and generate a precise candidate set as the top-k results.

## 4 Experiment

### 4.1 Experiment Settings

**Datasets.** We conduct experiments on 5 high-dimensional ANN benchmarks (See detailed description in Appendix D.1). Specifically, we evaluate *LIRA* on two small-scale datasets, SIFT [2] and GloVe [29]. We also show the scalability of *LIRA* on three large-scale datasets: Deep [3], BIGANN [17], and Yandex TI [40]. For the constraint in the RAM source, we subsample 50M data points for each large-scale dataset, following previous studies [12, 22].

**Baselines.** We evaluate *LIRA* as the meta index compared with four baselines, IVF in Faiss [18], IVFPQ [16], IVFFuzzy and BLISS [12](See detailed description in Appendix D.1). Specifically, we build the IVFFuzzy index to show the effectiveness of redundancy through centroid distance rank, where every data point is placed in the two nearest clusters. BLISS is a learning-to-index method that builds four groups of partitions, each with an independent model.

To simulate the practical two-level $k$NN search, we build two-level indexes, *LIRA*-HNSW, and evaluate the effectiveness of *LIRA* as the meta index among partitions. We build two-level indexes for baselines similarly. To exclude the effect of the internal indexes, we first evaluate the effectiveness of *LIRA* and baselines as the meta index in small-scale datasets and then use two-level indexes in large-scale datasets. The detailed experiments of convergence validation and sensitivity analysis are in Appendix D.2 and D.3.

**Settings.** In *LIRA*, the number of partitions $B$ is set as 64 and 1024 for small-scale and large-scale datasets, respectively. The $k$ of $k$NN is mainly set as 100 since we focus on addressing probing waste and long-tailed $k$NN distribution. In index construction, the redundancy percentage $\eta$ is set as 3% when using a meta index alone and is set as 100% when using a two-level index (See detailed sensitivity study on $\eta$ in Appendix D.3). In query process, we use threshold $\sigma$ to choose partitions with $\widehat{p^q} > \sigma$ as probing partitions and tune $\sigma$ from 0.1 to 1.0 with a step of 0.05.

The number of partitions of baselines is the same as *LIRA*. For BLISS, we follow the original setting, build four groups of partitions with four independent models in the index construction phase, and search for four groups of partitions in the query phase. For two-level indexes, the parameter of HNSW in graph building for limiting the edge of a data point is set as 32, and the search parameter of HNSW for limiting the length of the candidates set is set as 128.

**Evaluation metrics.** For the one-level meta index, we evaluate the performance of *LIRA* and baselines threefold: accuracy, efficiency, and query fan-out. (1) We use Recall@$k$ to evaluate the search accuracy. (2) We use the distance computations *cmp* (i.e., the total number of visited data points) in the probing partitions to indicate the search efficiency. (3) We record the *nprobe* to reflect the query fan-out and to reflect the effectiveness of partition pruning. For the

**Table 1: Performance at Recall@$k$=0.98 with various $k$.**

| $cmp$ | IVF | IVFPQ | IVFFuzzy | BLISS | *LIRA* |
|---|---|---|---|---|---|
| $k = 10$ | 120641 | 120641 recall=0.70 | 119409 (-1.0%) | 151911 (+25.9%) | **83824 (-30.5%)** |
| $k = 50$ | 137276 | 137276 recall=0.74 | 144120 (+4.9%) | 151911 (+10.6%) | **91431 (-33.3%)** |
| $k = 100$ | 137276 | 137276 recall=0.76 | 144120 (+4.9%) | 168778 (+22.9%) | **96261 (-29.8%)** |
| $k = 200$ | 153931 | 187410 recall=0.78 | 144120 (-6.3%) | 168778 (+9.6%) | **99279 (-35.5%)** |
| $nprobe$ | IVF | IVFPQ | IVFFuzzy | BLISS | *LIRA* |
| $k = 10$ | 7 | 7 recall=0.70 | **4 (-42.8%)** | 8 (+14.2%) | 4.8138 (-31.2%) |
| $k = 50$ | 8 | 8 recall=0.74 | **5 (-37.5%)** | 8 (0%) | 5.2342 (-34.5%) |
| $k = 100$ | 8 | 8 recall=0.76 | **5 (-37.5%)** | 9 (+12.5%) | 5.4648 (-31.6%) |
| $k = 200$ | 9 | 11 recall=0.78 | **5 (-44.4%)** | 10 (+11.1%) | 5.6561 (-37.1%) |

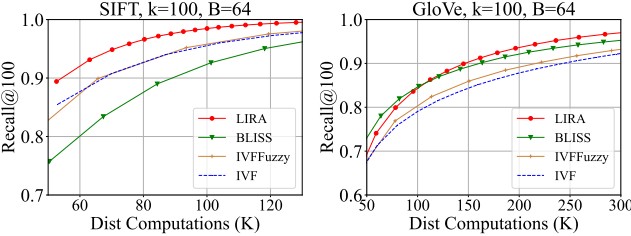

**Figure 7: Recall and $cmp$ on small-scale datasets.**

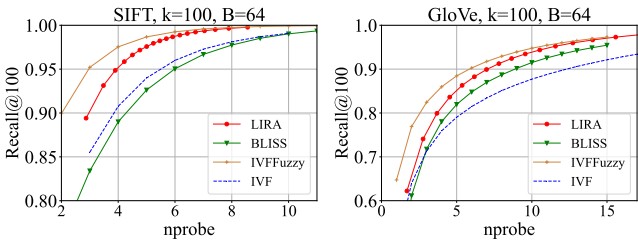

**Figure 8: Recall and $nprobe$ on small-scale datasets.**

two-level index in large-scale datasets, apart from the Recall@$k$ and $nprobe$, we additionally use query per second (QPS) to reflect the general search efficiency.

## 4.2 Evaluation on Small Scale Datasets

In this section, we first show the performance of *LIRA* and baselines when retrieving different $k$ nearest neighbors and illustrate the superiority of *LIRA* with large $k$ settings. Second, we show the trade-off between recall and distance computations and between recall and $nprobe$ on two small-scale datasets, SIFT and GloVe. For a fair comparison, we present the average distance computations executed by four individual models and partitions in BLISS.

**Performance with various $k$.** To explore the performance on different retrieval requirements, we conduct experiments on SIFT with various $k$ and calculate the minimum average distance computations and average $nprobe$ to achieve Recall@$k = 0.98$. When IVFPQ can hardly achieve the desired recall, we record the acceptable distance computations with the corresponding recall value.

Table 1 shows that the $cmp$ increases monotonically with $k$. Moreover, as $k$ increases with more serious long-tailed $k$NN distributions, the advantages of *LIRA* are gradually highlighted. There are two reasons behind such an advantage. First, utilizing the strength of adaptive probing from a query-aware view, *LIRA* focuses on eliminating the probing waste by directly probing target $k$NN partitions with the model rather than probing according to distance ranks to centroids. Second, *LIRA* reduces the long-tailed $k$NN distributions with reasonable replicas. Table 1 also shows the required $nprobe$ for various $k$, which gives an apparent advantage of *LIRA* in saving query fan-out. When *LIRA* and other baselines except IVFFuzzy achieve the same recall, *LIRA* needs less $nprobe$. This result supports that the probing model in *LIRA* can prune partitions effectively and accurately. Due to the strong advantage of *LIRA* in a large $k$, we mainly present the results for *Recall*@100 in the rest of the experiments for brevity. IVFFuzzy is superior in reducing $nprobe$, but it performs comparably to IVF on $cmp$, for the average partition size in IVFFuzzy is double than that of other baselines.

For the suboptimal performance of IVFPQ compared with IVF, we drop it from the remaining experiments.

**Trade-off between recall and both $nprobe$ and $cmp$.** To manipulate the recall with corresponding $cmp$, we tune the $nprobe$ in IVF, IVFFuzzy, and BLISS, and $\sigma$ for partition pruning in *LIRA*. As shown in Fig 7 and Fig. 8, *LIRA* surpasses all the baselines in *Recall*@100. (1) Compared with other partition-based methods, *LIRA* outperforms for two reasons. First, the well-built redundant partitions in *LIRA* can naturally reduce the optimal $nprobe^*$ and the probing quantity for partitions with long-tail $k$NN. (2) Second, the probing model can better adaptively narrow down the area of the probing partition with the well-learned mapping from a data point in the vector space to the $k$NN partitions. (3) Moreover, the gap between *LIRA* and the baselines expands as the recall increases on GloVe. This is because the baselines struggle to tackle the phenomenon of the long-tailed $k$NN distribution, and the long-tail data points mainly impact the trade-off between recall and search cost at a high recall value. Hence, the advantage of *LIRA* is more outstanding with a high recall requirement through an effective query-aware partitioning pruning strategy. (4) As for performance on SIFT, when recall is higher than 0.98, the gap between *LIRA* and the baselines slightly shrinks. This is because *LIRA* does not fully address the long-tailed $k$NN distribution for considering efficiency, resulting in some long-tail data points without duplication.

It is also worth noting that the learning-based method, BLISS, performs much worse than other methods on SIFT. There are two main reasons for the inefficiency of BLISS. (1) BLISS builds partitions with models from scratch instead of refining partitions based on other clustering methods, e.g., the K-Means clustering algorithm. The partitions built in BLISS tend to be unbalanced without fine-tuned training, resulting in some partitions with a large number of data points while some other partitions have no data points. These unbalanced partitions may lead to suboptimal partition probing and

**Table 2: Performance of QPS, Recall@100, partition pruning rate and overall latency on large-scale datasets.**

| Metrics | Query Per Second | | | Recall@100 | | | nprobe | | |
|---|---|---|---|---|---|---|---|---|---|
| Dataset | IVF | IVFFuzzy | *LIRA* | IVF | IVFFuzzy | *LIRA* | IVF | IVFFuzzy | *LIRA* |
| Deep | **126** | 93 | 117 | 0.8996 | 0.9604 | **0.9655** | **9** | 11 | 9.2586 |
| | 227 | 338 | **354** | 0.8265 | 0.8354 | **0.8441** | 5 | 3 | **2.9567** |
| BIGANN | 62 | 104 | **118** | 0.9464 | 0.9607 | **0.9639** | 19 | 11 | **10.1043** |
| | 173 | 230 | **257** | 0.8496 | 0.8860 | **0.8952** | 7 | 5 | **4.6465** |
| Yandex TI | 141 | 160 | **174** | 0.7673 | 0.8316 | **0.8386** | 9 | 8 | **7.7058** |
| | 258 | 324 | **346** | 0.6847 | 0.7524 | **0.7732** | 5 | 4 | **3.8171** |

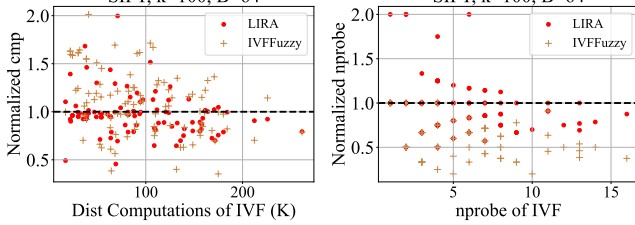

**Figure 9: Per query performance.**

cause more distance computations, *cmp*. (2) BLISS requires 4 groups of partition-based indexes. The *nprobe* of BLISS is presented as the fixed *nprobe* in one group of indexes, and the *cmp* of BLISS denotes the total number of candidates from four groups of partitions after deduplication. This also causes a waste of *cmp* since the four indexes built by independent models differ.

**Search performance per query.** Fig. 9 presents the normalized distance computations *cmp* and *nprobe* of IVFFuzzy and *LIRA* over IVF on a per-query basis, respectively. Take the figure on the left of Fig. 9 as an example for a detailed explanation. The x-axis denotes the minimum *cmp* when IVF achieves the *Recall@k* = 0.98 for every query. The y-axis presents the ratio between the *cmp* of another method to achieve a recall of 0.98 and the *cmp* of IVF. A red point below the normalized value of 1.0 infers that the query process of this query with *LIRA* takes less retrieval cost than IVF, and a brown plus sign infers the normalized cost of IVFFuzzy. We remove the baseline BLISS from this experiment because it struggles to achieve the target recall efficiently. This figure provides the performance of individual queries, where we sample 100 queries from all the queries of the dataset for display.

Overall, IVFFuzzy and *LIRA* reduce the *cmp* compared with IVF at the same recall. For IVF, there are easy queries and hard queries, where hard queries require more probing partitions and distance computation to achieve the target recall. (1) It is worth mentioning in Fig. 9, *LIRA* optimizes most of the queries that need *nprobe* ⩾ 10 with IVF. This means that *LIRA* exhibits a significant reduction, especially on hard queries. This is because there are more long-tailed *k*NN distributions for hard queries, which results in more probing waste. (2) For the IVFFuzzy, it is desired to achieve half of IVF's *nprobe* to achieve a comparable search efficiency in the one-level meta index, because the number of data points contained in one partition of IVFFuzzy is doubled compared to that of IVF. However, the general normalized *nprobe* of IVFFuzzy is more than 0.5. This illustrates that the redundancy strategy in IVFFuzzy is

unable to achieve comparable search efficiency when IVFFuzzy is used as a meta index alone.

## 4.3 Evaluation on Large Scale Datasets

This section presents the performance of *LIRA* and other baselines on three large-scale datasets. We build two-level indexes with IVF, IVFFuzzy and *LIRA* as the meta index, respectively. The HNSW index is used as the internal index for a fast inner-partition searching process. We drop BLISS from evaluation on large-scale datasets for out-of-memory with 4 groups of two-level indexes. Following previous study [12], Table 2 shows the performance in two scenarios that demand high efficiency or high recall, respectively. Based on the experiment results, we can make the following observations.

- Compared with non-learning methods, *LIRA* outperforms IVF and IVFFuzzy in most cases, especially with high recall.
- Compared to the performance of a one-level index, the IVF-Fuzzy becomes more efficient among the two-level indexes. With HNSW as the internal index, search efficiency is enhanced largely by avoiding exhaustive searches within a partition, and the large number of redundant data points has less impact on efficiency.
- Due to the learning-based redundancy and the query-aware adaptive *nprobe* generated by the effective probing model, *LIRA* can achieve better partition pruning than the IVF and IVFFuzzy.
- The improvement of *LIRA* compared to IVFFuzzy varies among different datasets, which may demonstrate that different $\eta$ is required for different datasets. Hence, an opportunity exists for *LIRA* to achieve a better redundancy with an adaptive number of redundant data points on different datasets.

## 5 Conclusion

State-of-the-art partition-based ANN methods typically divide the dataset into partitions and use query-to-centroid distance rankings for search. However, they have limitations in probing waste and long-tailed *k*NN distribution across partitions, which adversely affects search accuracy and efficiency. To overcome these issues, we propose *LIRA*, a LearnIng-based queRy-aware pArtition framework. Specifically, we propose a probing model to achieve outstanding partition pruning by reducing probing waste and providing query-dependent *nprobe*. Moreover, we introduce a learning-based redundancy strategy that utilizes the probing model to efficiently build redundant partitions, thereby mitigating the effects of long-tailed *k*NN distribution. Our proposed method exhibits superior performance compared with existing partition-based approaches in the accuracy, latency, and query fan-out trade-offs.

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

## A  Related Work

Existing studies for ANN search can be roughly divided into four groups, including (1) hash-based [14, 32], (2) tree-based [7, 13, 28], (3) quantization-based [1, 9, 25], and (4) graph-based [8, 24, 27, 36]. Typically, the computational cost of retrieval $k$ approximate nearest neighbors, $O(nd)$, incurs from the number of visited vectors denoted as $n$ and the dimension of vectors represented as $d$. The existing ANN search methods leverage high-dimensional indexes to reduce latency from these two aspects.

### A.1  Partition-based ANN methods

The tree-based indexes [7] partition the vector space into nested nodes and then narrow down the search area with hierarchical tree-based indexes during the search phase. DB-LSH et.al [32] efficiently generate candidates by dynamically constructing query-based search areas. IVF (inverted file) index [18] first clusters the vectors into partitions and then narrows down the search area with the nearest partitions. Chen et.al [5] uses a clustering algorithm and the inverted index to build balanced posting lists, and probes the clusters within a certain distance from the query vector. Zhang et.al [43] focuses on multi-probe ANN search and formalizing the query-independent optimization as a knapsack problem. Neural LSH [6] generates partitions by balanced graph partitioning. Zhao et.al [45] combine the strength of LSH-based and graph-based methods and utilize LSH to provide a high-quality entry point for searching in graphs.

### A.2  Learning-based ANN methods

Artificial Intelligence(AI) has been applied to databases [20, 35], as well as information retrieval systems [21]. We provide some learning-based ANN on top of partitions and graphs. For partition-based methods, BLISS [12] and Li et.al [22] combine the partition step and the learning step with learning-to-index methodology, and then search with a fixed $nprobe$. Zheng et.al [46] builds hierarchical balanced clusters and further leverages neural networks to generate adaptive $nprobe$ for each query. For graph-based methods, the graph-based indexes [27, 36] first connect the similar vectors with basic proximity graphs in the construction phase and then route in the graph through the most similar neighbors with the greedy search strategy. Based on the graph-based HNSW, Li et.al [19] demonstrate that easy queries often need less search depth than hard ones. Li et.al [19] introduces an early termination strategy and uses models to predict the minimum number of visited vectors required for retrieving the ground truth $k$NN for a given query, which can halt the search before meeting the traditional termination condition in the graph.

## B  More Details of Motivation

We find that probing according to $nprobe^*_{dist}$ wastes probing cardinality, with empirical study presented in Fig. 10 (LEFT). The blue dashed line shows the percentage of queries with $nprobe^*$ no more than a specified $nprobe$. The red dashed line reflects the percentage of queries with $nprobe^*_{dist}$ no more than a given $nprobe$. In other words, those queries achieving $Recall@10 = 1$ satisfy the accuracy requirement. (1) As we can see, the $nprobe^*_{dist}$ exceeds 20 for some queries when retrieving top-10 $k$NN, which introduces significant probing wastes. (2) Moreover, the waste of probing is even worse with a larger value of $k$. As shown in Fig. 10 (RIGHT) for $k = 100$, the $nprobe^*$ of all queries are no more than 22; while $nprobe^*_{dist}$ escalates to 40 for some queries.

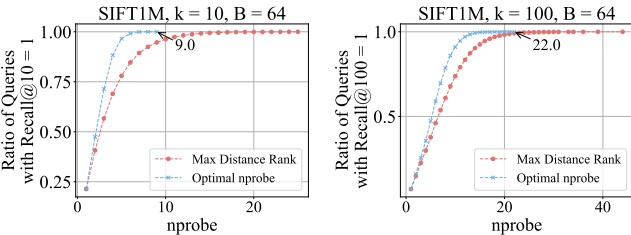

**Figure 10: Probing Waste with Distance Ranking and Common Long-tail Phenomenon.**

---

**Algorithm 1** One-level and Two-level Index Building

---

1: Input: data points $\mathcal{D}$, training queries $Q$
2: Output: indexes of $\mathcal{D}$ in $B$ partitions for top-$k$ retrieval
3: *(1) For one-level meta index*
4: Sample a subset of data $\mathcal{D}_{sub}$ for training     ▷ Scalability
5: Build $B$ partitions for $\mathcal{D}_{sub}$ and get partition centroids and distance to centroids $I$
6: Get $k$NN partition distributions $p$ of $\mathcal{D}_{sub}$ as labels
7: Learn $p^q$, $f(q, I) = p^q$     ▷ Model Training
8: Put all $\mathcal{D}$ in the nearest partitions
9: Get the predicted probability $\widehat{p}$ of $\mathcal{D}$ with $f(\cdot)$
10: Pick $\eta\%$ of data points as $\mathcal{D}_{pick}$ with $\widehat{p}$     ▷ Redundancy
11: **for all** data point $v$ in $\mathcal{D}_{pick}$ **do**
12:     Choose a target replica partition with $\widehat{p}$
13:     Duplicate the data of $v$
14: **end for**
15: *(2) For two-level index*
16: Build internal indexes for each partition

---

## C  More Details of Scalability of *LIRA*

Since the subset of data is used in training, we give a detailed explanation of the scalability of *LIRA*. In general, *LIRA* only requires the ground truth $k$NN and $k$NN count distribution of a subset of data. The two phases of probing model training and learning-based redundancy are both scalable to large-scale datasets, even if the model is trained on a subset. The algorithm of index building with the probing model is illustrated in Algorithm 1.

First, for model training, the true label of $k$NN partition distribution is more sparse if we scale the number of data with the same total number of partitions $B$. This is because a partition contains more data points with a large-scale dataset under the same $B$, and

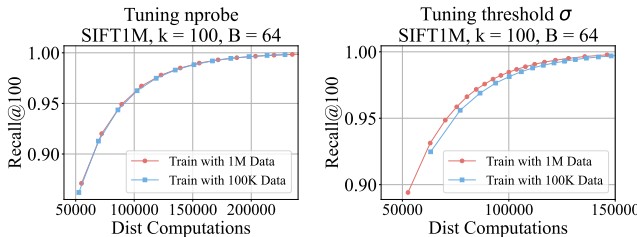

Figure 12: Scalability of model training.

Table 3: Datasets

| Dataset | # of Dimension | # of Data | # of Query |
|---------|---------------|-----------|------------|
| SIFT | 128 | 1M | 10K |
| GloVe | 96 | 1M | 1K |
| Deep | 96 | 50M | 10K |
| BIGANN | 128 | 50M | 10K |
| Yandex TI | 200 | 50M | 10K |

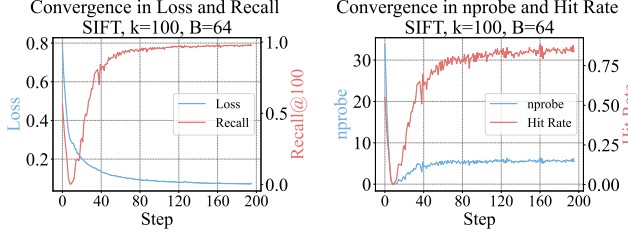

Figure 11: Model convergence validation with SIFT128, 1M.

the $k$NN of a query are more likely to be separated in fewer partitions. However, the true label of $k$NN partition distribution $p$ does not affect the model training, since the probing partitions are selected by the threshold $\sigma$ of the predicted possibilities.

Second, for learning-based redundancy, the picking and duplicating steps are all based on the relative results. We pick a data point $v$ with a relatively high quantity of predicted probing partitions and then duplicate it to a partition with relatively high probing possibility in $p^v$. Hence, the workflow of $LIRA$ is scalable to large-scale datasets.

# D More Details of Experiment

## D.1 Experiment Settings

**Datasets.** We conduct experiments on 5 high-dimensional ANN benchmarks with different data sizes and distributions. The details of the datasets are shown in Table. 3.

**Baselines.** The detailed information on baselines is as follows.

- IVF. IVFFlat in Faiss [18] (abbreviated as IVF) is a widely used ANN method that utilizes inverted indexes.
- IVFPQ. IVFPQ [16] is a widely adopted solution that combines the advantages of product quantization and the inverted file index.

- IVFFuzzy. Fuzzy clustering is a method where each data point can be put in more than one cluster. We build the IVFFuzzy index to show the effectiveness of redundancy through centroid distance rank, where every data point is placed in the two nearest clusters.
- BLISS [12]. BLISS is a learning-to-index method that builds groups of partitions, each with an independent model. The variant of BLISS [22] is omitted from experiments, as well as Neural LSH [6] that is inferior to BLISS.

**Evaluation Platform.** We implement our methods and baselines in Python 3.7. All the experiments are conducted on Intel(R) Xeon(R) Silver 4214 CPU @ 2.20GHz, 256GB memory, and 4 NVIDIA GeForce RTX 3080.

## D.2 Convergence Validation

This section is not intended to compare $LIRA$ against other baselines but rather to verify the convergence of $LIRA$. We illustrate that during the process of model training and re-partition, the probing model can achieve convergence while the recall and probing fan-out can also be improved simultaneously. All the experiments on convergence validation are conducted on SIFT with 10K queries in the setting of $k = 100$ and $B = 64$. The batch size for model training is set as 512. To evaluate whether the predicted partitions are the $k$nn partitions, we also calculate the hit rate of $k$nn partitions. We record the model inference time and the total search time for all queries, and we find that the model inference time only occupies less than 1% of the total search time, which shows the efficiency of predicting probing partitions.

**Loss and recall.** As shown in Fig. 11(LEFT), we record the loss and the recall of the testing queries during the training process. A step in the x-axis means every 10 batches of training data. We can observe that the loss in the blue line dramatically decreases and can finally achieve convergence. In addition, the recall in the red line decreases and then increases. This is because the positive label of target $k$NN partitions is sparse, and the probing model tends to predict few probing partitions at the beginning of training. With the probing model learning the mapping of data points to the target $k$NN partitions, the recall can approach nearly 1.0 at the end of the training.

***nprobe* and hit rate**. With the threshold of model outputs set as default $\sigma = 0.5$, we record the average *nprobe* of queries to represent the query fan-out and the hit rate of target probing partitions during the training process. As we can see in Fig. 11(RIGHT), the number of predicted *nprobe* in the blue line converges in stable and approximates the *nprobe*∗, and the hit rate in the red line can reach a high level. Hence, the result supports that the probing model can predict the target $k$NN partitions well. Even if the hit rate is about 0.8 after training, we can tune the threshold $\sigma$ in the query process. In detail, a less $\sigma$ results in more probing partitions and a larger *nprobe*, while a higher $\sigma$ works in the opposite.

**Scalability**. We evaluate the scalability of $LIRA$ by training on the subset of data and on the whole data, respectively. Specifically, for training on a subset, we sample $D_{Sub}$, 100K data from the 1M data in SIFT as the training data and building partitions on $D_{Sub}$. For $v \in D_{Sub}$, we get the $k$NN count distribution, $n^v$, and then get the $k$NN partition distribution $p^v$ in $D_{Sub}$ as training label. After model training, the learning-based redundancy is used on the whole data $D$. Keeping the partitions centroids unchanged, we put the whole

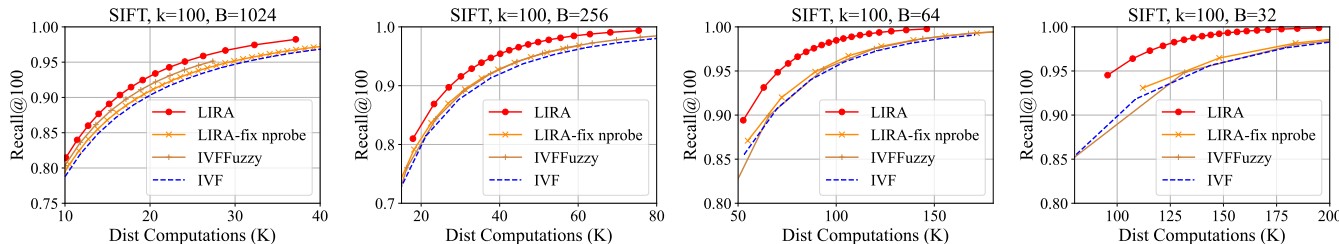

**Figure 13: Sensitivity of the number of partitions $B$ in SIFT.**

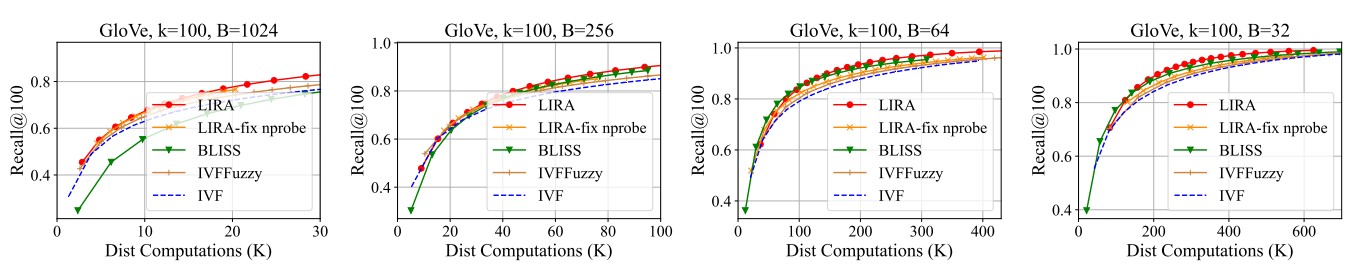

**Figure 14: Sensitivity of the number of partitions $B$ in GloVe.**

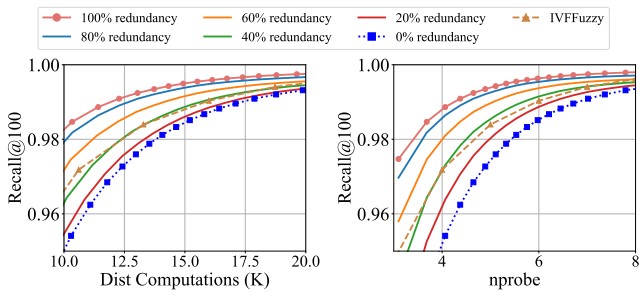

**Figure 15: Sensitivity of the redundancy ratio $\eta$ in SIFT.**

data $D$ in partitions and then use the probing model to achieve redundancy. For comparison, we also train a model with all the 1M data as training data.

We get the trade-off between the recall and distance computations by tuning the *nprobe* (i.e., probing partitions in the top *nprobe* output rank) and by tuning the threshold $\sigma$ (i.e., probing partitions with $\widehat{p^q} > \sigma$). As shown in Fig. 12, the model trained on a subset performs similarly to the model trained on the whole dataset whether probing with *nprobe* or threshold. The result supports that *LIRA* can be trained on a subset but can still achieve good partition redundancy and partition pruning on a whole dataset.

### D.3 Sensitivity Analysis

**Effect of $B$.** The hyper-parameter $B$ in *LIRA* is used as the total number of partitions. We conduct sensitivity analysis of the partition number $B$ on SIFT and GloVe, which is presented in Fig. 13 and Fig. 14, respectively. To reflect the general trade-off between efficiency and accuracy, we plot these two figures with the average distance computations versus recall, comparing *LIRA*-fix *nprobe*,

BLISS, IVFFuzzy, and IVF. The *LIRA*-fix *nprobe* is a variant of *LIRA* with different partition pruning, which utilizes a *nprobe* configuration instead of adaptive *nprobe* controlled by the threshold $\sigma$ of the model output. We set the partition number as $B \in \{1024, 256, 64, 32\}$. Due to the inefficiency of BLISS on SIFT in Fig. 7 and Fig. 8, we exclude the plotting of this group of settings for the result unable to fit in the graph.

In all settings considered, *LIRA* outperformed other methods. (1) For a small value of $B$, *LIRA* can reduce distance computations since one partition contains more data points, and reducing one partition probing can save many data points from searching. (2) For a large value of $B$, *LIRA* can also decrease distance computations because the probing waste of baselines is more severe across a large number of partitions. (3) For *LIRA*-fix *nprobe*, even if this variant method searches with *nprobe* configuration similar to the search process of IVF, it outperforms IVF consistently. This result supports that the probing model in *LIRA* can better directly probe the $k$NN partitions with model output rank than probe along with the centroid distance rank in IVF.

**Effect of $\eta$.** The hyper-parameter $\eta$ in *LIRA* is used as the redundancy ratio, picking the data points with the highest predicted *nprobe* for redundancy. Considering the internal index can reduce the adverse impact of redundancy of both *LIRA*-HNSW and IVFFuzzy-HNSW, we conduct sensitivity analysis of the redundancy ratio $\eta$ on SIFT, which is presented in Fig. 15. We tune the $\eta$ to duplicate different ratios of data points in *LIRA*-HNSW, while the IVFFuzzy-HNSW duplicates all data points. As we can see, merely duplicating 40% of data points in *LIRA*-HNSW can achieve comparable performance to IVFFuzzy-HNSW on the trade-off between recall and both distance computations and *nprobe*. For fair redundancy, we set the $\eta$ as 100% in *LIRA*-HNSW as the same redundancy in IVFFuzzy-HNSW. When testing meta index alone, we set $\eta$ as 3% in *LIRA* to keep the high efficiency.

