# OpenReview forum: "LIRA: A Learning-based Query-aware Partition Framework for Large-scale ANN Search"
_ACM.org/TheWebConf/2025/Conference — WWW 2025 Poster_

### Official Review · Reviewer_7Zh9 · 2024-11-19

**Novelty:** 5
**Technical Quality:** 6

**Review:**

This paper proposes LIRA (Learning-based Query-aware Partition Framework), an innovative learning-driven framework designed to optimize the efficiency and accuracy of large-scale Approximate Nearest Neighbor (ANN) search. By introducing a learning-based redundancy strategy and a query-aware probing model, LIRA effectively reduces partition probing redundancy in queries and mitigates the long-tail effect. Experimental results show that LIRA can significantly reduce query latency and fan-out, achieving over a 30% improvement compared to other methods. In terms of design and implementation, the LIRA framework offers new insights for enhancing performance in large-scale ANN retrieval.

**Questions:**

1.The inclusion of more advanced deep learning-based ANN methods, such as graph embedding methods, as baselines could provide a more comprehensive comparison of the system's performance.
2.Ablation studies on key components, such as the learning-based redundancy strategy and the query-aware probing model, should be conducted to quantify their impact on the overall performance.

**Reviewer Confidence:**

4: The reviewer is certain that the evaluation is correct and very familiar with the relevant literature

**Scope:**

4: The work is relevant to the Web and to the track, and is of broad interest to the community

---

### Official Review · Reviewer_yMoB · 2024-11-27

**Novelty:** 5
**Technical Quality:** 4

**Review:**

Approximate Nearest Neighbor Search (ANNS) over partition-based indices is a significant research topic in the current vector data management community. This paper aims to optimize the offline partitioning process and the online probing process using learning-based methods. This direction is valuable, as demonstrated by previous research. The paper is well-structured, and some example analyses are helpful. However, there are several issues the authors need to address.

**Questions:**

Q1. Some previous learning-based partitioning or probing methods (such as [43], [46]) are excluded from the experimental evaluation. Why are they not compared?

Q2. What is the offline preprocessing time? Does the proposed method require more offline preprocessing time compared to current methods?

Q3. Predicting nprobe also requires additional online query processing. What proportion does this constitute of the total query time?

Q4. In Figure 7, the trade-off between recall and distance computations may be unfair, as the distance computations metric does not cover the complete online overhead, especially for the proposed method in this paper.

Q5. More ablation studies should be conducted to evaluate the individual modules, such as the method for selecting data points to replicate, the strategy for duplicating data points, and the proposed query probing method.

**Reviewer Confidence:**

4: The reviewer is certain that the evaluation is correct and very familiar with the relevant literature

**Scope:**

4: The work is relevant to the Web and to the track, and is of broad interest to the community

---

### Official Review · Reviewer_RCju · 2024-12-01

**Novelty:** 6
**Technical Quality:** 6

**Review:**

#### Summary

This paper proposes LIRA, a learning-based query-aware partition framework for ANN search, to tackle the irrelevant partition probing and long-tailed knn distribution problems of existing tasks. Specifically, LIRA first indexes data points into knn partitions and marks the potential long-tailed samples. Then, it expands the hard partitions by duplicating the selected long-tail samples. Finally, LIRA retrieves the top partitions for each query.

#### Strengths

1. Well-motivated. The motivation is illustrated with plenty of details, making claims credible.
2. Clear presentation. The paper is well-structured and easy to follow.
3. Abundant experiments. Comprehensive results are provided to illustrate the effectiveness, especially scale-up evaluation on large datasets and efficiency test provided.

#### Weakness

Hard to find due to my limited domain knowledge.

**Questions:**

None

**Reviewer Confidence:**

2: The reviewer is willing to defend the evaluation, but it is likely that the reviewer did not understand parts of the paper

**Scope:**

4: The work is relevant to the Web and to the track, and is of broad interest to the community

---

### Official Review · Reviewer_F4pS · 2024-12-02

**Novelty:** 4
**Technical Quality:** 5

**Review:**

This paper introduces LIRA, a novel framework aimed at addressing critical challenges in approximate nearest neighbor (ANN) search. By integrating machine learning models with redundancy strategies, LIRA achieves notable improvements over existing partition-based approaches in terms of accuracy, latency, and query fan-out. However, the paper has several areas for improvement:

1. Motivation Needs Further Strengthening: The long-tailed distribution of 𝑘NN, a key aspect of the problem addressed, requires more comprehensive explanation and justification.

2. Insufficient Theoretical Analysis: The paper lacks a detailed analysis of the algorithm's complexity, which is crucial for understanding its scalability and practical implications.

3. Technical Presentation Could Be Enhanced: The technical sections of the paper would benefit from clearer explanations and more structured presentation to improve readability and comprehension.

4. Limited Novelty: While the framework is promising, the paper's contribution in terms of novelty appears constrained, as it builds primarily on existing methods with incremental improvements.

**Questions:**

Q1. In the introduction, it would be beneficial to include a real-world case study to illustrate the two limitations highlighted by the authors. This would provide concrete context and make the identified issues more relatable and compelling to readers.

Q2. What is the time complexity of building and searching with LIRA? The authors should provide a detailed comparison of LIRA's computational cost with SoTA methods to evaluate its efficiency comprehensively.

Q3. LIRA relies on additional data for training a model. How can the authors justify that this overhead is acceptable compared to non-learning-based methods? A discussion quantifying this trade-off would strengthen the argument.

Q4. In Section 3.3, the example provided for selecting data points is difficult to follow. Clarifying the explanation with more intuitive illustrations or a step-by-step walkthrough would significantly improve its comprehensibility.

**Reviewer Confidence:**

4: The reviewer is certain that the evaluation is correct and very familiar with the relevant literature

**Scope:**

4: The work is relevant to the Web and to the track, and is of broad interest to the community